# A self-operating broadband spectrometer on a droplet

P. Malara [1✉], A. Giorgini[1], S. Avino[1], V. Di Sarno[1], R. Aiello[1], P. Maddaloni[1], P. De Natale[2] & G. Gagliardi[1]

Small-scale Fourier transform spectrometers are rapidly revolutionizing infrared spectro-chemical analysis, enabling on-site and remote sensing applications that were hardly ima-ginable just few years ago. While most devices reported to date rely on advanced photonic integration technologies, here we demonstrate a miniaturization strategy which harnesses unforced mechanisms, such as the evaporation of a liquid droplet on a partially reflective substrate. Based on this principle, we describe a self-operating optofluidic spectrometer and the analysis method to retrieve consistent spectral information in spite of the intrinsically non-reproducible droplet formation and evaporation dynamics. We experimentally realize the device on the tip of an optical fiber and demonstrate quantitative measurements of gas absorption with a 2.6 nm resolution, in a 100 s acquisition time, over the 250 nm span allowed by our setup's components. A direct comparison with a commercial optical analyzer clearly points out that a simple evaporating droplet can be an efficient small-scale, inex-pensive spectrometer, competitive with the most advanced integrated photonic devices.

[1] Consiglio Nazionale delle Ricerche, Istituto Nazionale di Ottica (INO), via Campi Flegrei 34, 80078 Pozzuoli, Naples, Italy. [2] Consiglio Nazionale delle Ricerche, Istituto Nazionale di Ottica (INO), Largo E. Fermi 6, 50125 Firenze, Italy. ✉email: pietro.malara@ino.cnr.it

Detecting the spectral power distribution of electromagnetic radiation and how that is modified by the interaction with matter is one of the founding techniques of experimental science. Uncountable scientific and industrial applications require to identify/quantify specific compounds in a sample from its transmission (reflection) spectrum. Such a demand has indeed stimulated the development of an enormous variety of devices: scanning grating monochromators, optical spectrum analyzers, virtual image phase arrays as well as different types of Fourier transform spectrometers, each type with its own resolution, velocity, sensitivity and operating wavelength region. In recent years a large number of miniaturized spectrometers, in some cases even below the microscale[1,2], have been reported in the scientific literature. This trend, fueled by the advances of integrated photonics, is rapidly driving spectro-chemical analysis out of the laboratories, opening the way to on-site applications. Most miniaturized devices rely on one or more dispersive elements that split the incoming radiation into different spectral channels and a detector to measure the intensity at each channel[3–8]. An alternative is represented by Fourier Transform (FT) spectrometers, that employ an interferometer with a variable-length arm; as the Optical Pathlength Difference (OPD) between the arms changes, the interferometer returns an interferogram that is equivalent to the FT of the input radiation spectrum[9]. As opposed to dispersive spectrometers, FT-based devices have no direct correspondence between the recorded data points and the spectral channel. Such a non-sequential, multiplexed approach results in an enhanced signal-to-noise ratio (SNR), known as Fellgett advantage[10]. On the other hand, FT spectrometers require mechanical moving parts that represent a non-negligible issue for miniaturization. Micro-electro mechanical systems (MEMS) technology has first taken up the challenge, demonstrating in the last two decades several FT integrated interferometers on a chip scale[11–13]. Successively, a generation of FT microspectrometers with no moving parts has followed, to address the robustness issues connected with the mechanical tuning of MEMS. These devices, fabricated by e-beam lithography, mostly rely on non-mechanical electro/thermo-optic tuning[14–16], on the spatial sampling of a stationary-wave interferogram in a waveguide (known as SWIFTS spectrometers)[17,18], or employ multiple interferometers with fixed OPDs (SHS and RAFT spectrometers)[19–21]. These devices operate in most cases in the near infrared window, allowing to record spectra with resolutions ranging from 0.5 to 5 nm over a 100–500 nanometers span.

Here, we show that a simple droplet evaporating on the tip of a single-mode optical fiber can work as a broadband self-operating Fourier Transform spectrometer with spectral characteristics and robustness comparable to most advanced integrated photonic devices. The described system behaves as a scanning interferometer that employs the spontaneous displacement of an evaporating droplet's surface as a mechanical drive. Light directed towards the droplet through the fiber undergoes two partial reflections at the fiber-liquid and at the liquid–air interface (see Fig. 1b). The radiation backreflected by these boundaries partially couples back in the counterpropagating fiber mode and interfere. Now, while the position of the first boundary is fixed, the second recedes as the droplet evaporates. The phase-scan of the field reflected by such displacing surface generates a clearly detectable interferogram in the back-propagating modal intensity that codifies the whole history of the droplet's evaporation dynamics, allowing to reconstruct position and recession velocity of the liquid surface at any given moment of the evaporation process[22]. On the other hand, the described system is analogous to the scanning interferometers used in FT spectrum analyzers, therefore information on the spectral distribution of the radiation delivered to the droplet is in principle also encoded in the

backreflection signal. Extracting this information requires a strategy to counterbalance the non-reproducible nature of the droplet formation and its unpredictable evaporation dynamics, but allows to turn an extremely simple, disposable and zero-cost system into a robust, miniaturized Fourier transform spectrum analyzer. In the next sections we introduce the droplet spectrometer technique, discuss its features (resolution, bandwidth, recording time) and use an all fiber-optic setup to demonstrate its capability of analyzing the wavelength distribution of a broadband radiation source as well as performing quantitative spectrochemical analysis.

## Results

**Technique and experimental setup**. A droplet sitting on the connector ferrule of a single-mode optical fiber identifies two reflecting surfaces: the fiber-liquid interface and the liquid-air interface, with Fresnel reflectivities $R_1$ and $R_2$, respectively. If a radiation with spectral distribution $I(k)$ is delivered to the droplet through the fiber, the total intensity reflected into the back-propagating guided mode can be written as

$$I(L) = \int R_1 I(k)dk + \int C(L)R_2 T_1^2 I(k)dk$$
$$+ \int \sqrt{C(L)T_1^2 R_1 R_2} \cdot I(k)\cos(2kL)dk \tag{1}$$

Where $T_1=1-R_1$, $L$ is the droplet thickness, $0 < C(L) < 1$ is the modal coupling efficiency of the light backreflected in the fiber and integration is over the whole spectral span of the incident radiation.

As $L$ evolves monotonically from $L_0$ to $0$ throughout the evaporation process, the first two terms of Eq. (1) generate a slowly varying signal $I_{DC}(L)$. Actually, since the first term is comparatively negligible, $I_{DC}(L)$ practically coincides with the second term of Eq. (1) (see Supplementary Note 2). A fast component $I_{AC}(L)$ is instead generated by the third term of Eq. (1), which corresponds to the real part of the Fourier transform of the spectral distribution $I(k)$, except for the factor $\sqrt{C(L)T_1^2 R_1 R_2}$.

In the $I_{AC}(L)$ signal, $T_1^2$, $R_1$ and $R_2$ are constant factors determined by the optical properties of the liquid substance. $C(L)$ instead depends on the overlap between the droplet-reflected radiation and the fiber guided mode, and is therefore determined by the instantaneous shape of the liquid surface. Mitigating the effect of $C(L)$ is essential for the droplet-to-droplet reproducibility of the spectra. For this task one can exploit the independent information contained in $I_{AC}(L)$ and $I_{DC}(L)$. If the $C(L)$ factors in Eq. (1) can be brought out of their integration signs, they cancel out when considering the signal:

$$\sqrt{\frac{I_{DC}(0)}{I_{DC}(L)}}I_{AC}(L) \cong \int \sqrt{T_1^2 R_1 R_2} \cdot I(k)\cos(2kL)dk \tag{2}$$

A signal independent of the specific droplet shape can be thus obtained by combining $I_{AC}(L)$ and $I_{DC}(L)$, under the assumption that $C(L)$ is constant over the spectrum of the incident radiation. This signal deviates from the actual FT of the source spectral distribution $I(k)$ only by a material-dependent factor $\sqrt{T_1^2 R_1 R_2}$, that represents a constant spectrometer's response function. Details in supplementary Note 2.

The droplet spectrometer's experimental setup is illustrated in Fig. 1a. At each measurement, a small quantity (~2 µl) of 2-propanol alcohol is dropped on the ferrule of a 2.5 mm FC-PC fiber connector's (Fig. 1b). Isopropanol alcohol was selected for its negligible absorption and almost constant refractive index in the whole near infrared (NIR) region[23], which results in a flat response function. Upon deposition on the connector ferrule, the

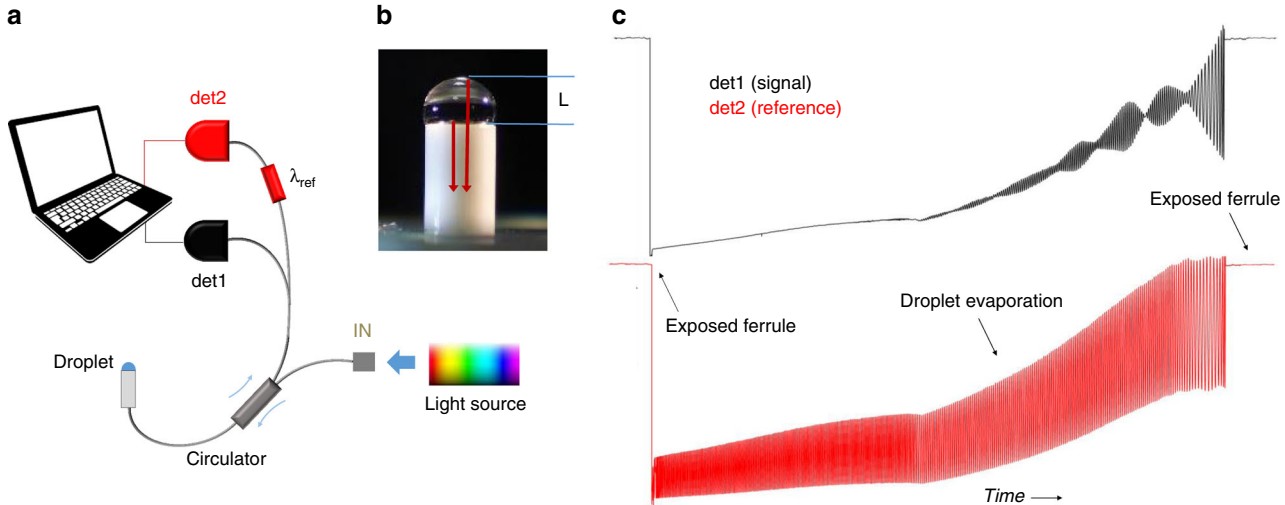

**Fig. 1 Experimental setup. a** Scheme of the fiber-optic droplet spectrometer; **b** detail of a droplet of thickness L sitting on the fiber connector's ferrule (water, for better visualization). The radiation reflected at the two droplet boundaries is indicated with red arrows; **c** signals from detectors det1 and det2 upon deposition and evaporation of an isopropyl alcohol droplet, obtained with a supercontinuum radiation source at the spectrometer input.

isopropanol droplet immediately shapes up and starts evaporating. The intensity backreflected into the fiber is collected through a fiber-optic circulator and sent to a InGaAs detector det1, except for a narrow spectral slice centered at $\lambda_{ref}$, that is tapped by a fiber-splitter and a notch filter and directed to a second photodiode det2.

Fig. 1c shows typical signals recorded with a broadband radiation at the spectrometer's input (specifically, a super-continuum source formed by mode-locked laser pulses spectrally broadened in a photonic fiber). Initially, the ferrule is exposed and the output levels of det1 and det2 (from now on referred to as "signal" and "reference" respectively) correspond to the Fresnel reflections of the input radiation at the fiber-air interface. When the liquid is deposited on the ferrule the detectors outputs abruptly drop; then, as a droplet forms and starts evaporating, they map the interference of the radiation fields backreflected into the optical fiber mode. When the droplet is completely evaporated, the connector ferrule is again exposed to air and both the detector outputs settle again to the initial level.

**Spectrum reconstruction**. To extract the radiation spectrum $I(k)$ with a Fourier Transform algorithm, the signal interferogram recorded in the time domain must be first transposed to a spatial scale, that is the instantaneous interferometer's optical path difference (OPD). For this task, the monochromatic reference signal is used. Indeed, in this interferogram, a full fringe corresponds to an OPD scan of one reference wavelength ($2n\Delta L=\lambda_{ref}$) and the last point before the full-evaporation plateau corresponds to OPD=0. In order to assign an instantaneous OPD coordinate to the $p$th point of this reference interferogram it is thus sufficient to count all the complete fringes from $p$ to the last point and assess the fraction of the fringe in which $p$ sits. The OPD axis of the reference interferogram can be directly transferred to the signal, because the two datasets are perfectly synchronous. However, since the evaporation velocity fluctuates unpredictably, the equally-spaced data points of the time-domain interferogram do not convert into equally-spaced data points in the OPD domain. Before applying the Fourier transform algorithm, the OPD-converted signal dataset must be thus interpolated with a continuous function and re-sampled at a constant rate. Additional details in supplementary Note 1.

In Fig. 2a, the interferograms extracted from the signals of Fig. 1c are shown at the end of the analysis procedure:

time-reversal, removal of the droplet-shape factors, transposition in the length domain and resampling. In Fig. 2b, the spectrum of the supercontinuum source, extracted by Fourier transform of the processed signal interferogram of panel a, is displayed as a red line and compared with a spectrum from a commercial optical analyzer (OSA, model ANDO AQ6317B) (black dashed curve). To better appreciate the capability of the droplet to reproduce all the spectral features of the radiation, both datasets were normalized to their max value. The excellent superposition of the two spectra validates the hypotheses leading to Eq. (2) in the wavelength range here considered, and demonstrates that a droplet can be an effective optical spectrum analyzer.

The spectral resolution $dk = \mathrm{OPD}_{max}^{-1} = \frac{1}{2nL_0}$ ($L_0$ is the initial droplet thickness) depends on the overall length of the recorded interferogram. With isopropanol on a standard 2.5 mm-diameter ferrule, it was possible to consistently record interferograms longer than 600 fringes at $\lambda_{ref} = 1538$ nm, which corresponds to a resolution $dk = 11$ cm$^{-1}$ (~2.6 nm) in a 2-min recording time. The nature of the liquid and the interfacial forces with the ferrule material determine the droplet shape, its initial thickness $L_0$ and its evaporation velocity. Indeed, depending on the wetting of the ferrule surface, the droplet evaporation can lead to a different spectral resolution and recording time. For example, as shown in Fig. 1b, water forms a droplet with $L_0 = 2$ mm on the fiber ferrule, thus allowing a spectral resolution $dk=2$ cm$^{-1}$ (~0.5 nm); however, such a large droplet takes a few tens of minutes to evaporate completely (water vapor pressure at the lab's temperature is 17.5 torr against the 48 torr of isopropanol). It is important that the liquid absorption is negligible in the wavelength span of interest, otherwise the accumulated heat may destabilize the droplet and cause its fragmentation in the late stages of the evaporation process.

**Absorption spectroscopy**. In a vast number of applications, spectrometers are employed to investigate the transmission (reflection) spectra of material samples. For this task, the spectrum of the radiation transmitted (reflected) by the sample is usually recorded and normalized by a reference spectrum of the interrogating source. It is worth noting that in this kind of measurement the presence of a spectrometer's response function is irrelevant (because it's automatically canceled by the normalization). On the other hand, because the two spectra are recorded

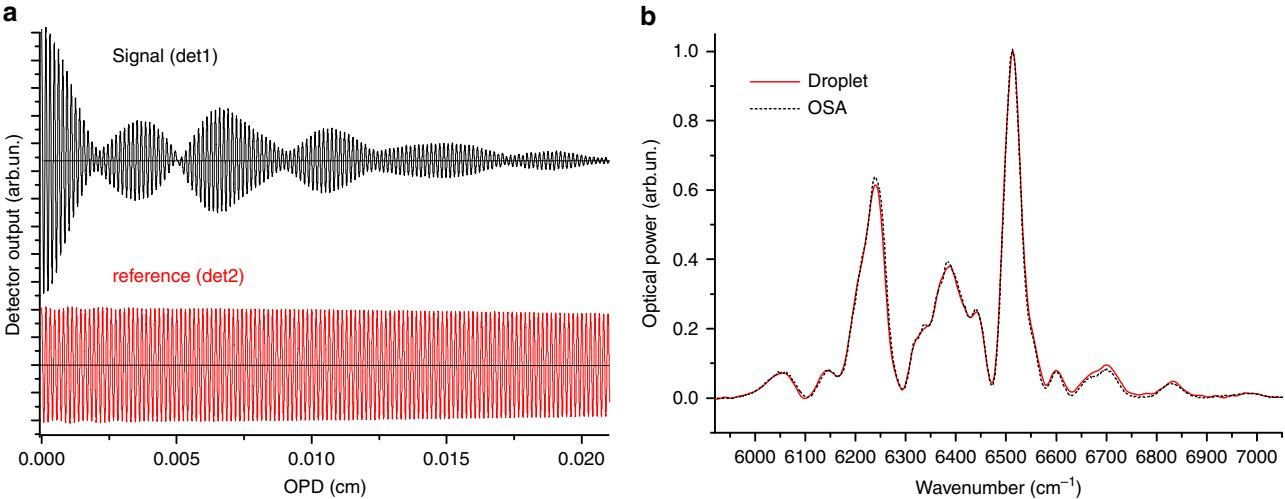

**Fig. 2 Spectral analysis of a light source. a** interferograms obtained from the raw signals of Fig. 1c after processing as in Eq. (2). For ease of visualization, the x scale span is limited to 0.02 cm; **b** red line: spectrum of a supercontinuum source obtained by discrete Fourier transform of the det1 interferogram (zero-filling factor=4, no apodization, no phase-correction, Resolution=11 cm$^{-1}$); black dashed line: same spectrum recorded with a commercial optical analyzer (8.5 cm$^{-1}$ resolution, 3-points adjacent averaging).

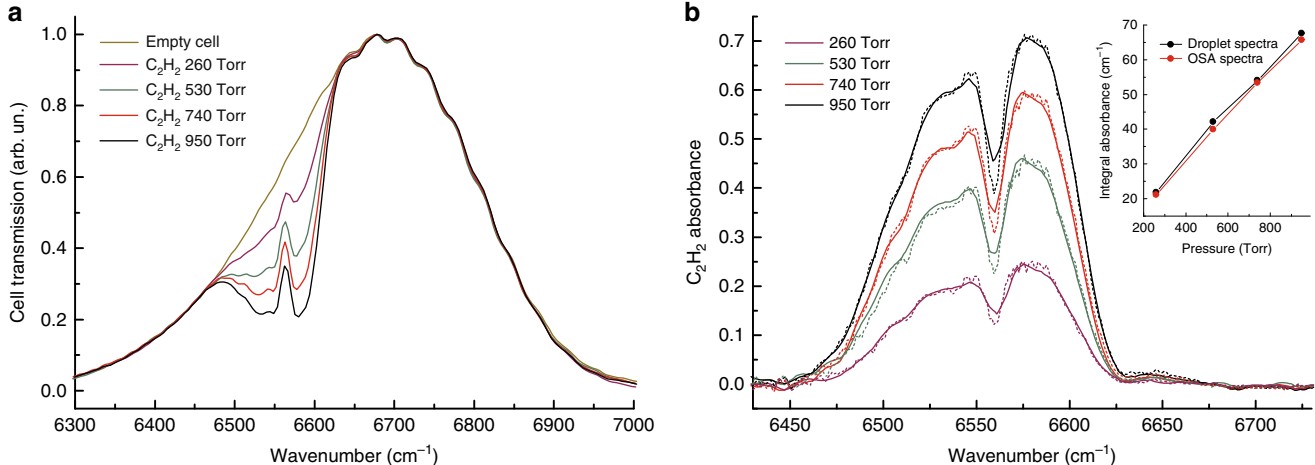

**Fig. 3 Acetylene absorption. a** transmission spectrum of a spontaneous emission source through an acetylene cell at different pressures (droplet spectrometer, Fourier transform settings as in the spectrum of Fig. 2); **b** acetylene absorbance curves obtained by off-line normalization of the spectra shown in panel **a** to the empty-cell transmission; dashed lines: same absorbance curves as measured by a commercial optical spectrum analyzer (8.5 cm$^{-1}$ resolution, 3-points adjacent averaging); inset: integral absorbance values calculated from the described spectra.

at different times, measuring the absorbance spectrum of a sample represents a severe reproducibility test for a spectrometer.

In Fig. 3, we show our results for quantitative detection of acetylene absorption. For these measurements, an incoherent radiation source (the spontaneous emission of a semiconductor optical amplifier) was sent to a 15-cm long acetylene gas cell equipped with inlet, outlet and pressure gauge. The cell transmission was analyzed both with the droplet spectrometer and the OSA for different values of the gas pressure. The droplet spectrum, shown in Fig. 3a, clearly shows the bell-shaped curve of the spontaneous emission source partially blocked by the absorption of acetylene at different pressures. In Fig. 3b, the acetylene absorbance curves are retrieved by normalization to the source radiation spectrum (recorded once and for all with the empty cell). In the same figure, the absorbance curves obtained with the spectrum analyzer (with a slightly higher resolution) are plotted as dashed lines. A more quantitative

comparison is also shown in the inset of Fig. 3b, where the integrated absorbances calculated from both the droplet and the OSA spectra are plotted. The shown ability to reproduce correctly the shape of the acetylene combination band using off-line normalization demonstrates the consistency of spectra obtained with different droplets.

## Discussion

A droplet evaporating on the tip of a fiber connector is analogous to a scanning-arm interferometer and can be therefore used to retrieve the spectrum of a radiation delivered into the fiber. Devising a strategy to cope with the non-reproducible nature of the evaporation process, we demonstrated that a simple droplet can accurately analyze the complex spectrum of a super-continuum radiation source and assess quantitatively the absor-bance of a gas sample in the near infrared region. In our experimental demonstration, isopropanol on a 2.5 mm fiber-optic

connector's ferrule allowed to retrieve 2.6 nm-resolution spectra in about 100 s (that could be much reduced by simply using a heat source to accelerate the evaporation). Different liquids can be used to obtain a different spectral resolution or recording time. The responsivity of the InGaAs detectors and the fiber circulator's bandwidth set the operating range of our demonstrative setup in the 6000–7000 cm$^{-1}$ interval (~250 nm span), but with appropriate detectors and fiber-optic equipment[24] such window can be easily extended to the whole NIR region or moved to the mid infrared.

The concept here demonstrated can be translated into an entire class of optofluidic analyzers, whereby evaporation or capillary forces provide the mechanical drive of a Fourier Transform spectrometer. A vast number of devices can be envisioned, ranging from broadly-accessible, almost zero–cost spectrometers to more complex and versatile arrangements, whereby integrated microfluidics and engineering of the ferrule surface allow to tune the liquid displacement dynamics (and thus the output spectral parameters) to fit specific applications.

## Data availability
Raw data that support the findings of this study are available from the corresponding author upon reasonable request.

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

## Acknowledgements
The authors wish to thank G. Notariale for the technical help and to acknowledge the stimulating discussions with Chema Senra on the evaporation dynamics of liquid droplets.

## Author contributions
PM conceived of the presented technique and designed the data analysis procedure. AG, and SA contributed to the experimental setup. VdS, RA and PMad contributed to the spectroscopic demonstration. PM, PdN and GG supervised the findings of this work. All authors discussed the results and contributed to the final manuscript.

## Competing interests
The authors declare no competing interests.
