## [Peer Review File · Nature Communications]

Reviewers' comments:

Reviewer #2 (Remarks to the Author):

A self-operating broadband spectrometer on a droplet

In this article, the authors describe an original optical-fiber-and-droplet-based small-scale Fourier-transform spectrometer (FTS). In the proposed design, the light to be analyzed (either before or after it interacts with the sample to be probed), is guided through a monomode optical fiber to the other where sits a liquid droplet. The light partially reflected at the fiber-liquid interface interferes (back in the fiber) with the light partially reflected at the upper surface of the droplet; interferences are monitored as a function of time with a photodiode working at its own detection rate. In this design, the natural process of evaporation of the droplet elegantly achieves the optical path difference (OPD) scan of the FTS, achieved in a conventional (Michelson-type) interferometer by motor-driven scan of the length of the reference arm. Spectral analysis (I as a function of wavenumber) of the light is retrieved after Fourier-transform of the recorded interferogram (I as a function of OPD). Resolution is inversely proportional to the maximal thickness of the droplet. In the given example (i.e. for the chosen ferrule's material and size as well as for the chosen liquid and volume), 2.6nm ($\sim 11 \text{ cm}^{-1}$) resolution over 250nm ($\sim 1000 \text{ cm}^{-1}$) span in the near-IR (1.5 μm) is achieved in 2 minutes, i.e. in the time needed for the isopropanol droplet to evaporate fully. A new droplet is needed to record a new spectrum.

Besides the very nice idea of using natural droplet evaporation as the driving force of the OPD scan, the authors devised an elegant and robust self-calibration of the instantaneous $\text{OPD} = 2nL(t)$, where L is the thickness of the droplet at all time, in order to compensate for the droplet-to-droplet non-reproducibility of evaporation kinetics. For doing so, they monitor (simultaneously to the main whole signal) the $I_{\text{ref}}(t)$ signal due to a single reference wavelength (isolated from the global signal thanks to a fiber splitter and the reflection from a notch filter). Instantaneous droplet thickness is then retrieved/calibrated knowing that (1) at the exact time of full evaporation L is null and I_{ref} is maximum (top of a fringe) and (2) that each full fringe corresponds to a full reference wavelength. The instantaneous OPD calibration is crucial and done in an accurate way, as a clock for their system, fixing all variations so that the trace becomes regular, as illustrated for both a supercontinuum and a SOA light source, with and without an absorbing gaseous sample in presence. The retrieved FT-spectra (as a function of wavelength) match indeed very nicely the spectra measured with a commercial dispersive-element-based spectrometer of slightly better spectral resolution (8.5 cm^{-1}). Current design is yet limited to a certain bandwidth, and depositing (manually) a new droplet on the fiber tip for each measurement is not yet very practical for field applications. However, these limitations could technically be overcome, as suggested by the authors, provided the spectra are not affected by the curvature of the droplet. Controlling droplet geometry and displacement dynamics should enable controlling spectral parameters according to targeted applications.

To my knowledge the paper covers new original material. It is very well written which makes it easily intelligible. The authors explained clearly their procedure and provide strong evidence of their conclusions. This paper definitely opens new perspective in the field of mechanics-free miniaturized FTS. I cannot estimate how important it would become but I believe that this paper will be the source of new work in the area and I think that the results should definitely be published, with only optional minor revisions in the manuscript and mandatory minor revisions in supplementary material.

LIST OF MINOR REVISIONS

1) Manuscript

- Abstract
- Line 17: I would suggest adding the time scale for recording one full spectrum (less than 2 minutes in the presented example)
- Experimental setup and results

- Line 68: I would suggest specifying a monomode fiber is used
- Line 73: I would suggest "L is the thickness of the droplet (or the position of the droplet's surface, that evolves monotonically from L_0 to 0 throughout evaporation process), $0 < C(L) \leq 1$ is the modal coupling efficiency of light back into the fiber and integration is over the whole spectral span of the incident radiation."
- Line 76: Add a coma before respectively.
- Lines 76-77: A clear definition of what is later called DC would be needed here (although explained in the supplementary information), and mention of the constant offset being neglected in front of the "slowly-varying" term in the following equation.
- Line 86: Eq.(3) should read Eq. (2)
- Figure 1.c: To ease the understanding of the manuscript at first sight I would suggest adding explicitly the name of the three phases "exposed ferrule", "Droplet evaporation" and "exposed ferrule" again as done in Fig. S1 of the supplementary material.
- Line 110: I would suggest specifying which type of supercontinuum source is used and why the spectrum isn't flat
- Line 118: each full fringe corresponds to a fixed optical path difference of one full wavelength
- Line 118: "before the full evaporation (DC) plateau"
- Line 119: I would suggest using another symbol than "n" to stand for the point number. Probably "p", like in the supplementary material.
- Line 120: replace integer "n" by "p"
- Line 121: replace integer "n" by "p"
- Line 129: I would mention here the type of supercontinuum source used
- Line 135: the author could potentially state that this validates their hypothesis of a wavelength-independent back coupling ($C(L)$) over the wavelength range tested.
- Figure 2: add (a) and (b) on the figure panels
- Line 150: delete extra "shape" before "generally"

- Absorption measurements

- Figure 3: add (a) and (b) on the figure panels
- Figure 3a: last label of the x axis is cut (it shows 690 instead of 6900 cm^{-1})
- Line 181: "same settings of fig2", but I believe that excludes the light source

2) General comments

Line 149, the authors note that liquids with larger surface tension generally shape up larger droplets. Actually, the size of the droplet depends primarily on the wetting properties of the surface of the ferrule. The surface can be hydrophilic, in which case the contact angle will be shallow and the droplet thin, or hydrophobic, in which case the droplet can be almost like a sphere and thick. The size of the bubble depends on the "contact angle". The authors could use this in their favor and treat the surface of the ferrule with a Silanization process to adjust the size of the bubble formed for a given liquid.

The author could potentially explain the limits of resolution imposed by the Nyquist criterium, which seem to make difficult to use a source at $1.5 \mu\text{m}$ to be the reference for an unknown signal at $1.5 \mu\text{m}$.

3) Supplementary material

- Line 13: suggestion: "the hereafter described operations 1) and 2) are performed"
- Line 23: uppercase at "Conversion"
- Line 27: replace "fringe" by "half-fringe". Refractive index is missing at the denominator in the whole paragraph
- Figure S2: add n at the denominator on figure and in caption
- Line 51: "2) Normalization..."
- Line 79: in Eq. (S2) a square root is missing at the denominator of the left-hand side of the equality
- Line 81: in Eq. (S3) the entire term square root of $DC(L)$ is missing
- Line 85: in Eq. (S4) the square root symbol is missing at the denominator of the central term (in

between the two equal symbols)

Reviewer #3 (Remarks to the Author):

In this paper, Malara et al. propose and demonstrate the use of an evaporating droplet at the tip of an optical fiber as a Fourier transform spectrometer with no moving part or time delay lines but with a spectral resolution sensitivity that has the potential to compete with the state of the art Fourier transform spectrometers based on advanced integrated photonic devices. The evaporating liquid droplet acts as a variable-length interferometer arm.

In the spectrometer demonstrated by Malara et al., light from a fiber is back reflected from two interfaces (fiber-liquid and liquid-air). One of the interfaces remains the same while the other is varying in time as the droplet evaporates. The interference of the light reflected from the two interfaces allows to reconstruct position and velocity of the liquid surface at any given moment of the evaporation process. This is the same principle a Fourier transform spectrometer with moving parts or delay lines is built on. By properly processing the recorded interferogram, the authors have succeeded to reproduce the spectrum of a supercontinuum source and the absorption spectrum of acetylene at various values of pressure.

This is an incredibly simple and low-cost hardware compared to optical spectrum analyzers and Fourier transform spectrometers. One may naively think that the irregular and non-reproducible evaporation velocity may be an issue in, for example, characterizing a radiation source, but the authors have showed that this can be overcome by analyzing simultaneously the interferograms obtained from the unknown radiation and a single-wavelength reference radiation.

I find this work to be very interesting. This is an elegant idea that will have immense potential in optofluidics and restricted-resource settings.

The manuscript is well-written; the authors have shown sufficient rigor in both the experiments in the main text and signal processing explained in the Supplement; the end-results and all the required information to understand the process is given; and the authors have interpreted the results and their implications properly. The motivation behind such a study is also justified.

In short, I find this paper to be very original and to have significant novelty to warrant publication in Nature Communications. Therefore, I recommend this work for publication.

There is a number of minor issues that needs to be addressed or highlighted in the revised manuscript to make the proposed idea and its pros & cons clearer.

Minor issues:

1. How will the irregularity of the droplet shape during the evaporation process affect the result? How do the authors deal with this in their system?
2. I understand that the authors may not have a state-of-the-art Fourier Transform spectrometer in their lab. I believe that there should be some reports in the literature about the acquisition time, resolution, sensitivity of these spectrometers. It will be good if the authors provide in a few sentences a comparison. Similarly, a comparison of what limits resolution, acquisition time and sensitivity in the conventional spectrometers and the one demonstrated in this work.
3. How will different liquids for droplets affect the outcomes? For example, what do authors expect to see if they use acetone instead of isopropanol?
4. Will it be better if the authors use an active evaporating scheme (e.g., using a heat source)

rather than let the evaporation takes place naturally?
5. How will the size of the droplet affect the outcome?

We sincerely wish to thank the reviewers for their careful reading of our manuscript. Their suggestions and to-the-point criticism gave us the opportunity to effectively improve the manuscript. In the following, we provide a point-by-point reply to their questions. The revisions described can be easily spotted in the texts with highlighted revisions (manuscript and supplementary material) following the replies sections.

Reviewer#2

In this article, the authors describe an original optical-fiber-and-droplet-based small-scale Fourier-transform spectrometer (FTS). In the proposed design, the light to be analyzed (either before or after it interacts with the sample to be probed), is guided through a monomode optical fiber to the other where sits a liquid droplet. The light partially reflected at the fiber-liquid interface interferes (back in the fiber) with the light partially reflected at the upper surface of the droplet; interferences are monitored as a function of time with a photodiode working at its own detection rate. In this design, the natural process of evaporation of the droplet elegantly achieves the optical path difference (OPD) scan of the FTS, achieved in a conventional (Michelson-type) interferometer by motor-driven scan of the length of the reference arm. Spectral analysis (I as a function of wavenumber) of the light is retrieved after Fourier-transform of the recorded interferogram (I as a function of OPD). Resolution is inversely proportional to the maximal thickness of the droplet. In the given example (i.e. for the chosen ferrule's material and size as well as for the chosen liquid and volume), 2.6nm (~ 11 cm⁻¹) resolution over 250nm (~ 1000 cm⁻¹) span in the near-IR (1.5 μ m) is achieved in 2 minutes, i.e. in the time needed for the isopropanol droplet to evaporate fully. A new droplet is needed to record a new spectrum.

Besides the very nice idea of using natural droplet evaporation as the driving force of the OPD scan, the authors devised an elegant and robust self-calibration of the instantaneous $OPD=2nL(t)$, where L is the thickness of the droplet at all time, in order to compensate for the droplet-to-droplet non-reproducibility of evaporation kinetics. For doing so, they monitor (simultaneously to the main whole signal) the $I_{ref}(t)$ signal due to a single reference wavelength (isolated from the global signal thanks to a fiber splitter and the reflection from a notch filter). Instantaneous droplet thickness is then retrieved/calibrated knowing that (1) at the exact time of full evaporation L is null and I_{ref} is maximum (top of a fringe) and (2) that each full fringe corresponds to a full reference wavelength. The instantaneous OPD calibration is crucial and done in an accurate way, as a clock for their system, fixing all variations so that the trace becomes

regular, as illustrated for both a supercontinuum and a SOA light source, with and without an absorbing gaseous sample in presence. The retrieved FT-spectra (as a function of wavelength) match indeed very nicely the spectra measured with a commercial dispersive-element-based spectrometer of slightly better spectral resolution (8.5 cm⁻¹). Current design is yet limited to a certain bandwidth, and depositing (manually) a new droplet on the fiber tip for each measurement is not yet very practical for field applications. However, these limitations could technically be overcome, as suggested by the authors, provided the spectra are not affected by the curvature of the droplet. Controlling droplet geometry and displacement dynamics should enable controlling spectral parameters according to targeted applications.

To my knowledge the paper covers new original material. It is very well written which makes it easily intelligible. The authors explained clearly their procedure and provide strong evidence of their conclusions. This paper definitely opens new perspective in the field of mechanics-free miniaturized FTS. I cannot estimate how important it would become but I believe that this paper will be the source of new work in the area and I think that the results should definitely be published, with only optional minor revisions in the manuscript and mandatory minor revisions in supplementary material.

Manuscript Abstract

- Line 17: I would suggest adding the time scale for recording one full spectrum (less than 2 minutes in the presented example)

The time needed for the droplet to evaporate completely is not reproducible as well, as it depends on the ambient temperature, amount of alcohol spilled in the vicinity of the ferrule, or whether evaporation is in a closed environment or not. For the measurement presented in this work it was around 100s, so we decided to adopt that value as reference of the recording time. We thus changed the claim in the abstract to "*...with 2.6 nm resolution, in a 100s acquisition time over the 250 nm span...*"

- Experimental setup and results - Line 68: I would suggest specifying a monomode fiber is used
We added this information in line 43 and in line 64.
- Line 73: I would suggest "L is the thickness of the droplet (or the position of the droplet's surface, that evolves monotonically from L₀ to 0 throughout evaporation

process), $0 < C(L) \leq 1$ is the modal coupling efficiency of light back into the fiber and integration is over the whole spectral span of the incident radiation.”

- Line 76: Add a coma before respectively.
- Lines 76-77: A clear definition of what is later called DC would be needed here (although explained in the supplementary information), and mention of the constant offset being neglected in front of the “slowly-varying” term in the following equation.

To organically address these suggestions, we rephrased the paragraphs from line 72 to 89. In the new version, we drop the definition AC, DC and define the slow and fast components $I_{DC}(L)$ and $I_{AC}(L)$ of the reflection signal. We explicitly mention that the constant offset originating from the first term can be neglected. The validity of such an approximation is now also shown in the supplementary materials.

- Line 86: Eq.(3) should read Eq. (2)
Corrected.
- Figure 1.c: To ease the understanding of the manuscript at first sight I would suggest adding explicitly the name of the three phases “exposed ferrule”, “Droplet evaporation” and “exposed ferrule” again as done in Fig. S1 of the supplementary material.

Done. In order to fit the whole exposed-evaporation-exposed transition of the signal, we had to rearrange some elements of fig.1.

Also, the figure caption and the figure discussion in the text (lines 107-114) have been revised accordingly.

Finally, now Fig.S1 of the supplementary materials becomes a duplicate of main text Fig.1c, so it has been removed.

- Line 110: I would suggest specifying which type of supercontinuum source is used and why the spectrum isn't flat

The strongly structured spectrum is typical of supercontinuum sources based on a mode-locked laser, i.e. consisting of mode-locked pulses spectrally broadened in a photonic crystal fiber. We now mention explicitly that our supercontinuum radiation is based on a mode-locked laser.

- Line 118: each full fringe corresponds to a fixed optical path difference of one full wavelength
 - Line 118: “before the full evaporation (DC) plateau”
 - Line 119: I would suggest using another symbol than “n” to stand for the point number. Probably “p”, like in the supplementary material.
 - Line 120: replace integer “n” by “p”
 - Line 121: replace integer “n” by “p”
 - Line 129: I would mention here the type of supercontinuum source used
- We revised the text from line 115 to 119 according to these suggestions.

- Line 135: the author could potentially state that this validates their hypothesis of a wavelength-independent back coupling (C(L)) over the wavelength range tested.
We agree: now we explicitly mention that the coincidence of the spectra shown in fig.2 validates the approximations leading to eq.(2). In the same spirit, at the end of the absorption measurement section, we now mention that the capability of recovering an absorbance spectrum by off-line normalization demonstrates the droplet-to-droplet reproducibility of the spectrometer.

- Figure 2: add (a) and (b) on the figure panels
Done

- Line 150: delete extra “shape” before “generally”
We removed this typo.

Absorption measurements

- Figure 3: add (a) and (b) on the figure panels
- Figure 3a: last label of the x axis is cut (it shows 690 instead of 6900 cm⁻¹)
Done
- Line 181: “same settings of fig2”, but I believe that excludes the light source
The word “settings” actually was just referring to the Fourier Transform algorithm and refers to zero filling, resolution, apodization, phase correction. To avoid the ambiguity pointed out by the reviewer, we changed the word in “Fourier transform settings”

General comments

- Line 149, the authors note that liquids with larger surface tension generally shape up larger droplets. Actually, the size of the droplet depends primarily on the wetting properties of the surface of the ferrule. The surface can be hydrophilic, in which case the contact angle will be shallow and the droplet thin, or hydrophobic, in which case the droplet can be almost like a sphere and thick. The size of the bubble depends on the “contact angle”. The authors could use this in their favor and treat the surface of the ferrule with a Silanization process to adjust the size of the bubble formed for a given liquid.

As correctly pointed out by the reviewer, the wettability of the surface, and thus the droplet shape, is determined by the balance between the cohesion force of the liquid (surface tension) and its adhesion to the surface. The possibility of using different liquids was suggested in different points of the original manuscript. We now grouped this scattered discussion in a dedicated paragraph (lines 145-152). The possibility of engineering the ferrule surface to control its wettability is also mentioned in the conclusive remarks (line 194)

- The author could potentially explain the limits of resolution imposed by the Nyquist criterium, which seem to make difficult to use a source at 1.5 μm to be the reference for an unknown signal at 1.5 μm .

The Nyquist theorem states that in order to correctly sample a signal that contains frequencies up to f , a minimum sampling resolution of $1/(2f)$ is necessary. In other words, at least two points per fringe are needed to sample a fringe.

In conventional FTIR spectrometers, the acquisition of the signal interferogram datapoints is triggered by the zero-crossing of a reference interferogram. In that case, the sampling rate depends on the wavelength of the reference beam (typically a He-Ne laser), and it would be indeed against the Nyquist criterium to sample a radiation shorter than 1.5 micron with a reference at 1.5 micron.

Instead, in our system the acquisition of datapoints is independent of the reference interferogram. It is only set by the data acquisition board. Considering that the interferograms trace roughly 700 fringes in two minutes (≈ 5 fringes per second), an average acquisition bandwidth of only 10 Hz (10 points per second) is sufficient to fulfill the Nyquist criterium. This means that even a cheap analog-to-digital converter allows plenty of oversampling.

The large oversampling of the reference interferogram, along with a synchronously recorded signal interferogram, allows to correctly sample signal spectral components even with a wavelength much shorter than the reference radiation.

This entire discussion has been also added to the supplementary materials (lines 40-53)

Supplementary material

After removing fig.S1 because equivalent to fig.1c of the main text, former figs S2, S3 and S4 are now S1, S2, S3.

- Line 13: suggestion: “the hereafter described operations 1) and 2) are performed”
- Line 23: uppercase at “Conversion”
- Line 27: replace “fringe” by “half-fringe”. Refractive index is missing at the denominator in the whole paragraph.
- Figure S2: add n at the denominator on figure and in caption

The observations are correct: we revised the text accordingly. As for the comment of line 27, the sentence now reads “*one fringe of the reference interferogram corresponds to a physical displacement of the droplet surface $\Delta L = \frac{\lambda_{ref}}{2n}$* ”

- Line 51: “2) Normalization...”
we renamed the subsections describing the analysis procedure. The names are now: “transposing the interferogram in the spatial domain” and “Making the interferogram independent of the droplet shape”
- Line 79: in Eq. (S2) a square root is missing at the denominator of the left-hand side of the equality
- Line 81: in Eq. (S3) the entire term square root of DC(L) is missing.
- Line 85: in Eq. (S4) the square root symbol is missing at the denominator of the central term (in between the two equal symbols)

The observations are correct: we revised the text accordingly. Also, the nomenclature AC, DC is now homogeneous to that of the manuscript: $I_{AC}(L)$, $I_{DC}(L)$.

Reviewer #3:

In this paper, Malara et al. propose and demonstrate the use of an evaporating droplet at the tip of an optical fiber as a Fourier transform spectrometer with no moving part or time delay lines but with a spectral resolution sensitivity that has the potential to compete with the state of the art Fourier transform spectrometers based on advanced integrated photonic devices. The evaporating liquid droplet acts as a variable-length interferometer arm.

In the spectrometer demonstrated by Malara et al., light from a fiber is back reflected from two interfaces (fiber-liquid and liquid-air). One of the interfaces remains the same while the other is varying in time as the droplet evaporates. The interference of the light reflected from the two interfaces allows to reconstruct position and velocity of the liquid surface at any given moment of the evaporation process. This is the same principle a Fourier transform spectrometer with moving parts or delay lines is built on. By properly processing the recorded interferogram, the authors have succeeded to reproduce the spectrum of a supercontinuum source and the absorption spectrum of acetylene at various values of pressure.

This is an incredibly simple and low-cost hardware compared to optical spectrum analyzers and Fourier transform spectrometers. One may naively think that the irregular and non-reproducible evaporation velocity may be an issue in, for example, characterizing a radiation source, but the authors have showed that this can be overcome by analyzing simultaneously the interferograms obtained from the unknown radiation and a single-wavelength reference radiation. I find this work to be very interesting. This is an elegant idea that will have immense potential in optofluidics and restricted-resource settings.

The manuscript is well-written; the authors have shown sufficient rigor in both the experiments in the main text and signal processing explained in the Supplement; the end-results and all the required information to understand the process is given; and the authors have interpreted the results and their implications properly. The motivation behind such a study is also justified.

In short, I find this paper to be very original and to have significant novelty to warrant publication in Nature Communications. Therefore, I recommend this work for publication.

There is a number of minor issues that needs to be addressed or highlighted in the revised manuscript to make the proposed idea and its pros & cons clearer.

Minor issues:

- How will the irregularity of the droplet shape during the evaporation process affect the result? How do the authors deal with this in their system?

As the droplet changes its shape during evaporation, its curved surface will inject more or less efficiently radiation back into the fiber, resulting in a modulation of the fringe amplitude described by the factor $C(L)$. Because the droplet shape is hardly predictable or controllable, this modulation will be different for each scan, hampering the droplet-to-droplet reproducibility of the spectra.

However, under the hypotheses that $C(L)$ is independent of the wavelength and that the first term of eq.(1) produces a negligible contribution to the signal, it is possible to combine the slow and the fast components of the recorded interferogram as in eq. (2), to remove the $C(L)$ dependency. The validity of these approximations is demonstrated by the coincidence of the droplet and the OSA spectra, shown in fig. 2b.

In the attempt to clarify any ambiguity on this important point, we took the following actions:

- a) We rephrased the lines from 72 to 119 in the manuscript, so that now it is more explicit that the analysis procedure is performed to deal with the droplet non reproducibility.
- b) We renamed the subsections describing the analysis procedure in the supplementary materials. They now read: “*transposing the interferogram in the spatial domain*” and “*Making the interferogram independent of the droplet shape*”

- I understand that the authors may not have a state-of-the-art Fourier Transform spectrometer in their lab. I believe that there should be some reports in the literature about the acquisition time, resolution, sensitivity of these spectrometers. It will be good if the authors provide in a few sentences a comparison. Similarly, a comparison of what limits resolution, acquisition time and sensitivity in the conventional spectrometers and the one demonstrated in this work.

We agree with the reviewer: our claim about the performance of our spectrometer being similar to that of other more complex devices can be more quantitative. On the other hand, we believe it would be useful to compare our spectrometer with other miniaturized devices demonstrated in literature, rather than with a state-of-the art macroscopic laboratory piece of equipment.

The performance of the miniaturized photonic spectrometers described in the introduction is quite variable: they range from the 5 nm resolution over 250 nm

demonstrated with the EO tuning interferometers, 5 nm resolution over 500 nm demonstrated by SWIFT spectrometers and 0.47 nm over 90 nm resolution demonstrated with RAFT spectrometers. These numbers well compare with the 2.5 nm resolution on a 250 nm span demonstrated in our work (also considering that our operating range is limited by the detectors and the optical fibers used in our setup).

To help the reader have a more quantitative idea of this comparison, in the revised text we introduce a sentence in lines 42-44 where we give an idea of the typical performance of the devices mentioned.

- **How will different liquids for droplets affect the outcomes? For example, what do authors expect to see if they use acetone instead of isopropanol?**

Depending on the wettability of the ferrule material with different liquids, taller or shorter droplets will form. Taller droplets return a higher resolution (because they set a longer optical path difference), but take more time to evaporate completely. So, there is typically a trade-off between resolution and recording time. In the revised manuscript version, the possibility of using different liquids is discussed explicitly in lines 145-152. However, not any liquid can be used. We did in fact notice that when using liquids that present a non-negligible absorption in the wavelength span of the incident radiation, the droplets tend to fragmentize in the last stages of evaporation, thus destroying the interferogram before reaching the zero optical path difference.

In the revised manuscript, we explicitly mention this effect in lines 150-152.

- **Will it be better if the authors use an active evaporating scheme (e.g., using a heat source) rather than let the evaporation takes place naturally?**

It is true that an active evaporation scheme could be a simple way to reduce the recording time. We now mention this possibility in line 189

- **How will the size of the droplet affect the outcome?**

See reply to point 3.

As a final remark, we added a line at the end of the text to acknowledge the precious discussions on droplet evaporation with dr C. Senra.

Also, we added a data availability statement after the references section.

A self-operating broadband spectrometer on a droplet

P. Malara^{1,*}, A. Giorgini¹, S. Avino¹, V. Di Sarno, R. Aiello, P. Maddaloni, P. De Natale² and G. Gagliardi¹.

¹ Consiglio Nazionale delle Ricerche, Istituto Nazionale di Ottica (INO), via Campi Flegrei, 34—Comprensorio A. Olivetti, 80078 Pozzuoli (Naples), Italy.

² Consiglio Nazionale delle Ricerche, Istituto Nazionale di Ottica (INO), Largo E. Fermi 6—50125 Firenze, Italy.

ABSTRACT: Small-scale Fourier transform spectrometers are rapidly revolutionizing infrared spectro-chemical analysis, enabling on-site and remote sensing applications that were hardly imaginable just few years ago. While most devices reported to date rely on advanced photonic integration technologies, here we demonstrate a miniaturization strategy which harnesses unforced mechanisms, such as the evaporation of a liquid droplet on a partially reflective substrate. Based on this principle, we describe a self-operating optofluidic spectrometer and the analysis method to retrieve consistent spectral information in spite of the intrinsically non-reproducible droplet formation and evaporation dynamics. We experimentally realize the device on the tip of an optical fiber and demonstrate quantitative measurements of gas absorption with a 2.6 nm resolution, in a 100s acquisition time, over the 250 nm span allowed by our setup's components. A direct comparison with a commercial optical analyzer clearly points out that a simple evaporating droplet can be an efficient small-scale, inexpensive spectrometer, competitive with the most advanced integrated photonic devices.

Introduction

Detecting the spectral power distribution of electromagnetic radiation and how that is modified by the interaction with matter is one of the founding techniques of experimental science. Uncountable scientific and industrial applications require to identify/quantify specific compounds in a sample from its transmission (reflection) spectrum. Such a demand has indeed stimulated the development of an enormous variety of devices: scanning grating monochromators, optical spectrum analyzers, virtual image phase arrays as well as different types of Fourier transform spectrometers, each type with its own resolution, velocity, sensitivity and operating wavelength region. In recent years a large number of miniaturized spectrometers, in some cases even below the microscale^{1,2}, have been reported in the scientific literature. This trend, fueled by the advances of integrated photonics, is rapidly driving spectro-chemical analysis out of the laboratories, opening the way to on-site applications. Most miniaturized devices rely on one or more dispersive elements that split the incoming radiation into different spectral channels and a detector to measure the intensity at each channel³⁻⁸. An alternative is represented by Fourier Transform (FT) spectrometers, that employ an interferometer with a variable-length arm; as the Optical Pathlength Difference (OPD) between the arms changes, the interferometer returns an interferogram that is equivalent to the FT of the input radiation spectrum⁹. As opposed to dispersive spectrometers, FT-based devices have no direct correspondence between the recorded data points and the

37 spectral channel. Such a non-sequential, multiplexed approach results in an enhanced signal-to-noise ratio
 38 (SNR), known as Fellgett advantage¹⁰. On the other hand, FT spectrometers require mechanical moving parts
 39 that represent a non-negligible issue for miniaturization. Micro-electro mechanical systems technology (MEMS)
 40 has first taken up the challenge, demonstrating in the last two decades several FT integrated interferometers on a
 41 chip scale¹¹⁻¹³. Successively, a generation of FT microspectrometers with no moving parts has followed, to
 42 address the robustness issues connected with the mechanical tuning of MEMS. These devices, fabricated by e-
 43 beam lithography, mostly rely on non-mechanical electro/thermo-optic tuning¹⁴⁻¹⁶, on the spatial sampling of a
 44 stationary-wave interferogram in a waveguide (known as SWIFTS spectrometers)^{17,18}, or employ multiple
 45 interferometers with fixed OPDs (SHS and RAFT spectrometers)¹⁹⁻²¹. These devices operate in most cases in the
 46 near infrared window, allowing to record spectra with resolutions ranging from 0.5 to 5nm over a 100-500
 47 nanometers span.

48 Here, we show that a simple droplet evaporating on the tip of a single-mode optical fiber can work as a
 49 broadband self-operating Fourier Transform spectrometer with spectral characteristics and robustness
 50 comparable to most advanced integrated photonic devices. The described system behaves as a scanning
 51 interferometer that employs the spontaneous displacement of an evaporating droplet's surface as a mechanical
 52 drive. Light directed towards the droplet through the fiber undergoes two partial reflections at the fiber-liquid
 53 and at the liquid-air interface (see fig.1b). The radiation backreflected by these boundaries partially couple back
 54 in the counterpropagating fiber mode and interfere. Now, while the position of the first boundary is fixed, the
 55 second recedes as the droplet evaporates. The phase-scan of the field reflected by such displacing surface
 56 generates a clearly detectable interferogram in the back-propagating modal intensity that codifies the whole
 57 history of the droplet's evaporation dynamics, allowing to reconstruct position and recession velocity of the
 58 liquid surface at any given moment of the evaporation process²². On the other hand, the described system is
 59 analogous to the scanning interferometers used in FT spectrum analyzers, therefore information on the spectral
 60 distribution of the radiation delivered to the droplet is in principle also encoded in the backreflection signal.
 61 Extracting this information requires a strategy to counterbalance the non-reproducible nature of the droplet
 62 formation and its unpredictable evaporation dynamics, but allows to turn an extremely simple, disposable and
 63 zero-cost system into a robust, miniaturized Fourier transform spectrum analyzer.

64 In the next sections we introduce the droplet spectrometer technique, discuss its features (resolution,
 65 bandwidth, recording time) and use an all fiber-optic setup to demonstrate its capability of analyzing the
 66 wavelength distribution of a broadband radiation source as well as performing quantitative spectrochemical
 67 analysis.

68

69 **Experimental setup and results.**

70 A droplet sitting on the connector ferrule of a single-mode optical fiber identifies two reflecting surfaces: the
 71 fiber-liquid interface and the liquid-air interface, with Fresnel reflectivities R_1 and R_2 , respectively. If a radiation
 72 with spectral distribution $I(k)$ is delivered to the droplet through the fiber, the total intensity reflected into the
 73 backpropagating guided mode can be written as

74

$$I(L) = \int R_1 I(k) dk + \int C(L) R_2 T_1^2 I(k) dk + \int \sqrt{C(L) T_1^2 R_1 R_2} \cdot I(k) \cos(2kL) dk \quad (1)$$

75

76

77 Where $T_i = 1 - R_i$, L is the droplet thickness, $0 < C(L) < 1$ is the modal coupling efficiency of the light backreflected
 78 in the fiber and integration is over the whole spectral span of the incident radiation.

79 As L evolves monotonically from L_0 to 0 throughout the evaporation process, the first two terms of eq.(1)
 80 generate a slowly varying signal $I_{DC}(L)$. Actually, since the first term is comparatively negligible, $I_{DC}(L)$
 81 practically coincides with the second term of eq.(1) (see suppl. Materials). A fast component $I_{AC}(L)$ is instead
 82 generated by the third term of eq.(1), which corresponds to the real part of the Fourier transform of the spectral
 83 distribution $I(k)$, except for the factor $\sqrt{C(L) T_1^2 R_1 R_2}$.

84 In the $I_{AC}(L)$ signal, T_1^2 , R_1 and R_2 are constant factors determined by the optical properties of the liquid
 85 substance. $C(L)$ instead depends on the overlap between the droplet-reflected radiation and the fiber guided
 86 mode, and is therefore determined by the instantaneous shape of the liquid surface. Mitigating the effect of $C(L)$
 87 is essential for the droplet-to-droplet reproducibility of the spectra. For this task one can exploit the independent
 88 information of contained in $I_{AC}(L)$ and $I_{DC}(L)$. If the $C(L)$ factors in eq.(1) can be brought out of their
 89 integration signs, they cancel out when considering the signal:

$$\sqrt{\frac{I_{DC}(0)}{I_{DC}(L)}} I_{AC}(L) \cong \int \sqrt{T_1^2 R_1 R_2} \cdot I(k) \cos(2kL) dk \quad (2)$$

91 A signal independent of the specific droplet shape can be thus obtained by combining $I_{AC}(L)$ and $I_{DC}(L)$, under
 92 the assumption that $C(L)$ is constant over the spectrum of the incident radiation. This signal deviates from the
 93 actual FT of the source spectral distribution $I(k)$ only by a material-dependent factor $\sqrt{T_1^2 R_1 R_2}$, that represents a
 94 constant spectrometer's response function. Additional details can be found in the supplementary materials.

95
 96 *Fig.1 –a) droplet spectrometer experimental setup b) detail of a droplet of thickness L on the connector's ferrule*
 97 *(water, for better visualization). The radiation reflected at the two droplet boundaries is indicated with red*
 98 *arrows; c) signals from detectors det1 and det2 following deposition and evaporation of an isopropyl alcohol*
 99 *droplet, obtained with a supercontinuum radiation source at the spectrometer input.*

100
 101

102 The droplet spectrometer's experimental setup is illustrated in fig.1a. At each measurement, a small quantity
 103 ($\square 2$ -ul) of 2-propanol alcohol is dropped on the ferrule of a 2.5mm FC-PC fiber connector's (detail in fig.1b).
 104 Isopropanol alcohol was selected for its negligible absorption and almost constant refractive index in the whole
 105 near infrared (NIR) region [23], which results in a flat response function. Upon deposition on the connector
 106 ferrule, the isopropanol droplet immediately shapes up and starts evaporating. The intensity backreflected into
 107 the fiber is collected through a fiber-optic circulator and sent to a InGaAs detector det1, except for a narrow
 108 spectral slice centered at λ_{ref} , that is tapped by a fiber-splitter and a notch filter and directed to a second
 109 photodiode det2.

110 Fig.1c shows typical signals recorded with a broadband radiation at the spectrometer's input
 111 (specifically, a supercontinuum source formed by mode-locked laser pulses spectrally broadened in a photonic
 112 fiber). When the liquid is placed on the exposed ferrule, the det2 and det1 outputs (from now on referred to as
 113 "signal" and "reference" respectively) immediately drop from the "exposed ferrule" level, that corresponds to
 114 the Fresnel reflection of the fiber-air interface. After a short transient, a droplet forms and the signals map the

115 polychromatic and monochromatic ($@\lambda_{ref}$ only) interference of the radiation fields reflected into the optical
116 fiber mode. The interference occurs until the droplet is completely evaporated and the connector is again
117 exposed to air, and both the detector outputs go back to their “exposed ferrule” level.

118 To extract the radiation spectrum $I(k)$ with a Fourier Transform algorithm, the signal interferogram
119 recorded in the time domain must be first transposed to a spatial scale, that is the instantaneous interferometer’s
120 optical path difference (OPD). For this task, the monochromatic reference signal is used. Indeed, in this
121 interferogram, a full fringe corresponds to an OPD scan of one reference wavelength ($2n\Delta L = \lambda_{ref}$) and the last
122 point before the full-evaporation plateau corresponds to $OPD=0$. In order to assign an instantaneous OPD
123 coordinate to the p -th point of this reference interferogram it is thus sufficient to count all the complete fringes
124 from p to the last point and assess the fraction of the fringe in which p sits. The OPD axis of the reference
125 interferogram can be directly transferred to the signal, because the two datasets are perfectly synchronous.
126 However, since the evaporation velocity fluctuates unpredictably, the equally-spaced data points of the time-
127 domain interferogram do not convert into equally-spaced data points in the OPD domain. Before applying the
128 Fourier transform algorithm, the OPD-converted signal dataset must be thus interpolated with a continuous
129 function and re-sampled at a constant rate. Details of the whole data analysis procedure can be found in the
130 supplementary materials.

131 In fig.2a, the interferograms extracted from the signals of fig.1c are shown at the end of the analysis
132 procedure: time-reversal, removal of the droplet-shape factors, transposition in the length domain and
133 resampling. In fig.2b, the spectrum of the supercontinuum source, extracted by Fourier transform of the
134 processed signal interferogram of panel (a), is displayed as a red line and compared with a spectrum from a
135 commercial optical analyzer (OSA, model ANDO AQ6317B) (black dashed curve). To better appreciate the
136 capability of the droplet to reproduce all the spectral features of the radiation, both datasets were normalized to
137 their max value. The excellent superposition of the two spectra validates the hypotheses leading to eq.(2) in the
138 wavelength range here considered, and demonstrates that a droplet can be an effective optical spectrum analyzer.

139
140
141 Fig 2 – (a) processed droplet interferograms from the raw signals of fig.1c. The x scale span is limited to 0.02 cm
142 for ease of visualization; (b) red line: spectrum of the supercontinuum source obtained by discrete Fourier
143 transform of the det1 interferogram (zero-filling factor=4, no apodization, no phase-correction, resolution=11
144 cm^{-1}); black dashed line: spectrum of the supercontinuum source recorded with a commercial optical analyzer
145 ($8.5cm^{-1}$ resolution, 3-points adjacent averaging).
146

147 The spectral resolution $dk = OPD_{max}^{-1} = \frac{1}{2nL_0}$ (L_0 is the initial droplet thickness) depends on the overall length
 148 of the recorded interferogram. With isopropanol on a standard 2.5mm-diameter ferrule, it was possible to
 149 consistently record interferograms longer than 600 fringes at $\lambda_{ref}=1538\text{nm}$, which corresponds to a resolution
 150 $dk=11\text{ cm}^{-1}$ ($\square 2.6\text{ nm}$) in a 2-min recording time. The nature of the liquid and the interfacial forces with the
 151 ferrule material determine the droplet shape, its initial thickness L_0 and its evaporation velocity. Indeed,
 152 depending on the wetting of the ferrule surface, the droplet evaporation can lead to a different spectral resolution
 153 and recording time. For example, as shown in fig.1b, water forms a droplet with $L_0=2\text{mm}$ on the fiber ferrule,
 154 thus allowing a spectral resolution $dk\square 2\text{cm}^{-1}$ ($\square 0.5\text{nm}$); however, such a large droplet takes a few tens of
 155 minutes to evaporate completely (water vapour pressure at the lab's temperature is 17.5 torr against the 48 torr of
 156 isopropanol). It is important that the liquid absorption is negligible in the wavelength span of interest, otherwise
 157 the accumulated heat may destabilize the droplet and cause its fragmentation in the late stages of the evaporation
 158 process.

160 **Absorption measurements.** In a vast number of applications, spectrometers are employed to investigate the
 161 transmission (reflection) spectra of material samples. For this task, the spectrum of the radiation transmitted
 162 (reflected) by the sample is usually recorded and normalized by a reference spectrum of the interrogating source.
 163 It is worth noting that in this kind of measurement the presence of a spectrometer's response function is
 164 irrelevant (because it's automatically cancelled by the normalization). On the other hand, because the two spectra
 165 are recorded at different times, measuring the absorbance spectrum of a sample represents a severe
 166 reproducibility test for a spectrometer.

167 In fig.3, we show our results for quantitative detection of acetylene absorption. For these measurements,
 168 an incoherent radiation source (the spontaneous emission of a semiconductor optical amplifier) was sent to a 15-
 169 cm long acetylene gas cell equipped with inlet, outlet and pressure gauge. The cell transmission was analyzed
 170 both with the droplet spectrometer and the OSA for different values of the gas pressure. The droplet spectrum,
 171 shown in fig.3a, clearly shows the bell-shaped curve of the spontaneous emission source partially blocked by the
 172 absorption of acetylene at different pressures. In fig.3b, the acetylene absorbance curves are retrieved by
 173 normalization to the source radiation spectrum (recorded once and for all with the empty cell). In the same
 174 figure, the absorbance curves obtained with the spectrum analyzer (with a slightly higher resolution) are plotted
 175 as dashed lines. A more quantitative comparison is also shown in the inset of fig.3b, where the integrated
 176 absorbances calculated from both the droplet and the OSA spectra are plotted. The shown ability to reproduce
 177 correctly the shape of the acetylene combination band using off-line normalization demonstrates the consistency
 178 of spectra obtained with different droplets.

179

180

181

182 Fig.3. (a) Transmission spectrum of a spontaneous emission source through an acetylene cell at different
183 pressures (droplet spectrometer, same **Fourier transform** settings of fig2); b) continuous lines: acetylene
184 absorbance curves at different pressures obtained by off-line normalization to the source spectrum; dashed lines:
185 acetylene absorbance curves as measured by a commercial optical spectrum analyzer (8.5 cm^{-1} resolution, 3-
186 points adjacent averaging).

187

188 **Discussion.** A droplet evaporating on the tip of a fiber connector is analogous to a scanning-arms interferometer
189 and can be therefore used to retrieve the spectrum of a radiation delivered into the fiber. Devising a strategy to
190 cope with the non-reproducible nature of the evaporation process, we demonstrated that a simple droplet can
191 accurately analyze the complex spectrum of a supercontinuum radiation source and assess quantitatively the
192 absorbance of a gas sample in the near infrared region. In our experimental demonstration, isopropanol on a
193 2.5mm fiber-optic connector's ferrule allowed to retrieve 2.6 nm-resolution spectra in $\sim 100\text{s}$ (that could be
194 much reduced by simply using a heat source to accelerate the evaporation). Different liquids can be used to
195 obtain a different spectral resolution or recording time. The responsivity of the InGaAs detectors and the fiber
196 circulator's bandwidth set the operating range of our demonstrative setup in the $6000\text{-}7000\text{ cm}^{-1}$ interval (~ 250
197 nm span), but with appropriate detectors and fiber-optic equipment²⁴ such window can be easily extended to the
198 whole NIR region or moved to the mid infrared.

199 The concept here demonstrated can be translated into an entire class of optofluidic analyzers, whereby
200 evaporation or capillary forces provide the mechanical drive of a Fourier Transform spectrometer. A vast number
201 of devices can be envisioned, ranging from broadly-accessible, almost zero-cost spectrometers to more complex
202 and versatile arrangements, whereby integrated microfluidics and **engineering of the ferrule surface** allow to tune
203 the liquid displacement dynamics (and thus the output spectral parameters) to fit specific applications.

204

205 **Acknowledgements**

206 The authors wish to acknowledge the stimulating discussions with C. Senra on the evaporation dynamics of
207 liquid droplets.

208

209

210 **References**

- 211 1. Yang, Z. *et al.* Single-nanowire spectrometers. *Science* **365**, 1017–1020 (2019).
- 212 2. Bao, J. & Bawendi, M. G. A colloidal quantum dot spectrometer. *Nature* **523**, 67–70 (2015).
- 213 3. Cheng, R. *et al.* Broadband on-chip single-photon spectrometer. *Nat. Commun.* **10**, 1–7 (2019).
- 214 4. Faraji-Dana, M. *et al.* Compact folded metasurface spectrometer. *Nat. Commun.* **9**, 1–8 (2018).
- 215 5. Jiang, A.-Q. *et al.* Ultrahigh-resolution spectrometer based on 19 integrated gratings. *Sci. Rep.*
216 **9**, 1–7 (2019).
- 217 6. Xia, Z. *et al.* High resolution on-chip spectroscopy based on miniaturized microdonut
218 resonators. *Opt. Express* **19**, 12356 (2011).
- 219 7. Feder, K. S., Westbrook, P. S., Ging, J., Reyes, P. I. & Carver, G. E. In-fiber spectrometer using
220 tilted fiber gratings. *IEEE Photonics Technol. Lett.* **15**, 933–935 (2003).
- 221 8. Wolfenbittel, R. F. State-of-the-art in integrated optical microspectrometers. *IEEE Trans.*
222 *Instrum. Meas.* **53**, 197–202 (2004).
- 223 9. Bell, R. *Introductory Fourier Transform Spectroscopy*. (Elsevier, 2012).
- 224 10. Fellgett, P. B. On the Ultimate Sensitivity and Practical Performance of Radiation Detectors.
225 *JOSA* **39**, 970–976 (1949).
- 226 11. Antila, J. *et al.* MEMS- and MOEMS-Based Near-Infrared Spectrometers. in *Encyclopedia of*
227 *Analytical Chemistry* 1–36 (American Cancer Society, 2014).
- 228 12. Erfan et al. - 2016 - On-Chip Micro-Electro-Mechanical System Fourier Tr.pdf.

- 229 13. Manzardo, O., Herzig, H. P., Marxer, C. R. & Rooij, N. F. de. Miniaturized time-scanning
230 Fourier transform spectrometer based on silicon technology. *Opt. Lett.* **24**, 1705–1707 (1999).
- 231 14. Souza, M. C. M. M., Grieco, A., Frateschi, N. C. & Fainman, Y. Fourier transform spectrometer
232 on silicon with thermo-optic non-linearity and dispersion correction. *Nat. Commun.* **9**, 1–8
233 (2018).
- 234 15. Chao, T.-H. *et al.* Compact liquid crystal waveguide based Fourier transform spectrometer for
235 in-situ and remote gas and chemical sensing. in (eds. Casasent, D. P. & Chao, T.-H.) 69770P
236 (2008).
- 237 16. Zheng, S. N. *et al.* Microring resonator-assisted Fourier transform spectrometer with enhanced
238 resolution and large bandwidth in single chip solution. *Nat. Commun.* **10**, (2019).
- 239 17. Le Coarer, E., Blaize, S., Benech, P., Stefanon, I., Morand, A., Le Rondel, G., Leblond, G.,
240 Kern, P., Fedeli, J.-M., Royer, P., “Wavelength-scale stationary-wave integrated Fourier
241 transform Spectrometry”, *Nature Photonics* (2007), 1, 8, 473 – 478).
- 242 18. Pohl, D. *et al.* An integrated broadband spectrometer on thin-film lithium niobate. *Nat.*
243 *Photonics* 1–6 (2019) doi:10.1038/s41566-019-0529-9.
- 244 19. Hugh Podmore, Alan Scott, Pavel Cheben, Aitor V. Velasco, Jens H. Schmid, Martin Vachon,
245 and Regina Lee, “Demonstration of a compressive-sensing Fourier-transform on-chip
246 spectrometer” *Optics Letters*, Vol. 42, Issue 7, pp. 1440-1443 (2017)
- 247 20. Nedeljkovic, M. *et al.* Mid-Infrared Silicon-on-Insulator Fourier-Transform Spectrometer Chip.
248 *IEEE Photonics Technol. Lett.* **28**, 528–531 (2016).
- 249 21. Kita, D. M. *et al.* High-performance and scalable on-chip digital Fourier transform
250 spectroscopy. *Nat. Commun.* **9**, 1–7 (2018).
- 251 22. Preter, E. *et al.* Fiber-Optic Evaporation Sensing: Monitoring Environmental Conditions and
252 Urinalysis. *J. Light. Technol.* **34**, 4486–4492 (2016).
- 253 23. Sani, E. & Dell’Oro, A. Spectral optical constants of ethanol and isopropanol from ultraviolet to
254 far infrared. *Opt. Mater.* **60**, 137–141 (2016)
- 255 24. Birks, T. A., Knight, J. C. & Russell, P. S. J. Endlessly single-mode photonic crystal fiber. *Opt.*
256 *Lett.* **22**, 961–963 (1997).

257
258
259 Raw data that support the findings of this study are available from the corresponding author, P. Malara,
260 upon reasonable request.

261
262
263
264
265
266
267
268
269
270

271
272
273
274
275
276
277

Supplementary informations for:

A self-operating broadband spectrometer on a droplet

P. Malara^{1,*}, A. Giorgini¹, S. Avino¹, V. Di Sarno, R. Aiello, P. Maddaloni, P. De Natale² and G. Gagliardi¹.

¹Consiglio Nazionale delle Ricerche, Istituto Nazionale di Ottica (INO), via Campi Flegrei, 34—Comprensorio A. Olivetti, 80078 Pozzuoli (Naples), Italy.

²Consiglio Nazionale delle Ricerche, Istituto Nazionale di Ottica (INO), Largo E. Fermi 6—50125 Firenze, Italy.

The interferograms recorded by det1 and det2 (displayed in fig.1c of the manuscript) consist of two arrays of N intensity values. The time spacing between successive interferogram datapoints is dt . A preliminary operation is to cut and reverse the arrays so that their first point corresponds to the instant when the liquid surface is totally evaporated. In order to extract the source spectrum by Fourier-transforming the recorded data, the hereafter described operations 1) and 2) are performed.

1) *Transposing the interferogram in the spatial domain*

To obtain the FT spectrum in wavenumbers, the recorded time-domain interferograms must be expressed in a length scale. For this task, the basic notions to consider are that point #1 of both interferograms corresponds to $L=0$ and one fringe of the reference interferogram corresponds to a physical displacement of the droplet surface $\Delta L = \frac{\lambda_{ref}}{2n}$. In the following procedure, first, we assign a length axis to the reference interferogram (det2):

- define an array ZDP with all the positions of the zero derivative points of the reference interferogram. The half-fringe spacing between two successive points corresponds to half a wavelength of the reference radiation.
- For each point p of the reference interferogram find the i -th element of the ZDP array such that $ZDP(i) < p < ZDP(i+1)$.
- assign to point p the coordinate $L(p) = \left(i - 1 + \frac{ZDP[i+1]-p}{ZDP[i+1]-ZDP[i]} \right) \frac{\lambda_{ref}}{2n}$, that represents the instantaneous physical distance between the droplet surface and the fiber ferrule. The integer number $(i-1)$ represents the number of half fringes between the first interferogram point and p (because the interferogram starts with a maximum, the first half fringe is completed at $i=2$); the second term describes the position of the point p as a fraction of the i -th half-fringe.

The length scale so obtained for the reference interferogram can be directly transferred to the signal interferogram. In fact, because the two signals are synchronous, the p -th point of the det2 interferogram has the same spatial coordinate of the p -th point of the det1 one. Once transposed in the length domain, the interferogram datapoints are not equally spaced (because they were recorded at a constant time separation dt , and evaporation is not constant in time). Last step to complete the x -scale data processing is therefore to interpolate the array, obtain a continuous function, then resample it with a fixed resolution. The det1 interferogram is then made of equally-spaced datapoints in the spatial domain.

31

32

33 Fig.S1: example of assignment of the coordinate $L(p)$ to the p -th datapoint. The numbers in green represent the element# of the ZPD
 34 array. In this example, for the p th point, $i=5$, so $L(p) = \left(4 + \frac{ZDP[6]-p}{ZDP[6]-ZDP[5]}\right) \frac{\lambda_{ref}}{2n}$

35

36

37 It is worth remarking how the data recording and sampling of the described system is different from that of conventional
 38 FTIR spectrometers. In the latter instruments, the acquisition of the signal datapoints is triggered by the zero-crossing of the
 39 reference interferogram, so in each period of the reference radiation 2 signal datapoints are recorded. According to the
 40 Nyquist theorem, this allows to correctly sample only spectral components with a longer period in the interferogram. i.e.
 41 those with an wavelength longer than the reference (typically an 633 nm He-Ne laser).

42 Instead, in our system, the acquisition rate of datapoints is only set by the data acquisition board. This means that even the
 43 cheapest analog-to-digital converter allows plenty of oversampling. Indeed, consider that the reference interferogram traces
 44 around 700 fringes in two 2 minutes (5 fringes per second in average). According to the Nyquist criterium sampling that
 45 would require an acquisition bandwidth of only 10 Hz (10 points per second). By largely oversampling the reference
 46 interferogram and synchronously recording the signal, it is possible to correctly sample also spectral components faster (i.e.
 47 with a wavelength much shorter) than the reference radiation.

48

49 2) Making the interferogram independent of the droplet shape

50 The signal interferogram, now in the length domain, can be written as :

$$I(L) = \int R_1 I(k) dk + \int C(L) R_2 T_1^2 I(k) dk + \int \sqrt{C(L) T_1^2 R_1 R_2} \cdot I(k) \cos(2kL) dk \quad (S1)$$

51 Eq. S1 is the sum of three contributions that from now on will be referred to as A, B and C. The first goal is to isolate the
 52 fast and the slow components $I_{AC}(L)$ and $I_{DC}(L)$ from the recorded interferogram.

53 $I_{DC}(L)$ is generated by the first two terms of eq. (S1): $I_{DC}(L) = A + B$. During the evaporation, these terms contribute
 54 to the signal with a fixed and a slowly varying offset respectively. The term A corresponds to the total intensity reflected by
 55 the fiber-liquid interface alone, and can be easily assessed from the backreflection of the fiber when completely immersed in
 56 the liquid. For any practical purpose, $A \ll B$ when using isopropanol, so $I_{DC}(L) \cong B$.

57 The slowly varying offset B can be extracted from the signal interferogram by detecting its zero derivative points and
 58 building the arrays: $max(i)$ and $min(i)$. The i -th array element is a 2-dimensional vector that maps position and signal level
 59 of the i -th maximum (minimum). The max and min arrays are then used to build the “MidPoints” array, where every point is
 60 defined as $MP(i) = \left\{ \frac{\min(i,1) - \max(i,1)}{2}, \max(i,2) - \frac{\max(i,2) - \min(i,2)}{2} \right\}$. Interpolation of the MidPoint array elements gives an
 61 experimental record of the term B(L), and it is shown as a red line in fig. S2. By subtracting it from the interferogram,
 62 $I_{AC}(L)$ is obtained (plotted in the inset of fig.S2).

63

64 Fig. S2: Extraction of the slow and fast signal components from the det1 interferogram: maxima and minima are detected (blue points) in
 65 order to build the Midpoint array (red curve), whose interpolation represents $I_{DC}(L)$ (in this example the procedure is applied before
 66 transposing to the length scale). Inset: $I_{AC}(L)$ retrieved by subtracting the slow component.

67

68 $I_{AC}(L)$ corresponds to the term C of equation (S1), so it is equivalent to the Fourier transform of the source spectrum except
 69 for the factor $\sqrt{C(L)T_1^2R_1R_2}$. Now, while T_1^2, R_1, R_2 depend only on the droplet constituent material, C(L) depends on the
 70 specific droplet shape and evaporation dynamics. In general, C increases as the droplet gets smaller, and produces a slowly
 71 increasing amplitude envelope in $I_{AC}(L)$ that is different from droplet to droplet.

72 It is possible to use the information contained in $I_{DC}(L)$ to remove the C(L) term dependence of $I_{AC}(L)$ before
 73 Fourier-transforming it. First step is to divide $I_{AC}(L)$ by the square root of $I_{DC}(L)$:

$$\frac{I_{AC}(L)}{\sqrt{I_{DC}(L)}} = \frac{\int \sqrt{C(L)T_1^2R_1R_2} \cdot I(k) \cos(2kL) dk}{\sqrt{\int C(L)R_2T_1^2I(k)dk}} \quad (S2)$$

74 In the hypothesis that C(L) does not depend on the wavenumber k, the terms $\sqrt{C(L)}$ in the numerator and denominator can
 75 be extracted from the integrals and cancel out. The new signal becomes:

$$\frac{I_{AC}(L)}{\sqrt{I_{DC}(L)}} = \frac{\int \sqrt{T_1^2R_1R_2} \cdot I(k) \cos(2kL) dk}{\sqrt{\int R_2T_1^2I(k)dk}} \quad (S3)$$

76 Now, the denominator represents the square root of the intensity reflected by the second surface into the fiber mode when
 77 C=1 (perfect coupling). This condition occurs at the very end of the evaporation process, when the droplet is just a thin
 78 layer, so all the intensity reflected by its outer boundary can be assumed to couple back into the fiber. The value
 79 $\int R_2T_1^2I(k)dk$ is thus just $I_{DC}(0)$. By multiplying for the square root of this factor we finally get:

$$I_{AC}(L) \sqrt{\frac{I_{DC}(0)}{I_{DC}(L)}} = \int \sqrt{T_1^2R_1R_2} \cdot I(k) \cos(2kL) dk \quad (S4)$$

80 The recorded signal so manipulated is now equal to the Fourier transform of the source spectrum except for the factor
 81 $\sqrt{T_1^2R_1R_2}$, that depends only on the droplet material and not on its specific shape or evaporation dynamics. This factor can
 82 be accounted for as a fixed response function of the spectrometer.

83

84

85 *Fig.S3: The very weak wavelength dependence of the factor $\sqrt{T_1^2 R_1 R_2}$ allows to approximate the Fourier transform of the*
 86 *AC2 signal to the spectrum of the radiation with an accuracy order of 1% across the whole NIR region.*

87

88 The response $\sqrt{T_1^2 R_1 R_2}$ for isopropanol is plotted in figS3 (rescaled to 1) as a function of wavelength. The refractive
 89 indexes $n_{iso}(\lambda)$ and $n_{air}(\lambda)$ of isopropanol and air were evaluated using Sellmeier equations [1,2] while the fiber effective
 90 refractive index $n_{fiber}(\lambda)$ was calculated starting from the group-index values reported in the specs sheet of the Corning
 91 SMF28 optical fiber. These dispersion curves allow to calculate the Fresnel reflectivities/transmissivities $R_1(\lambda)$, $R_2(\lambda)$ and
 92 $T_1^2(\lambda)$ and therefore the wavelength dependence of the spectrometer response function. The plot of fig.S3 clearly shows
 93 that, besides a scaling factor, the $\sqrt{T_1^2 R_1 R_2}$ is flat to the 1% tolerance in all the NIR window. The FT of the AC₂ (L)
 94 interferogram is therefore an accurate approximation of the actual spectrum of the radiation, as can be easily seen in fig.2b
 95 of the manuscript.

96

97

98 References

99 [1] P. E. Ciddor. "Refractive index of air: new equations for the visible and near infrared", Appl. Optics 35, 1566-1573
 100 (1996)

101 [2] E. Sani, A. Dell'Oro, "Spectral optical constants of ethanol and isopropanol from ultraviolet to far infrared". *Opt.*
 102 *Mater.* **60**, 137–141 (2016)

103

REVIEWERS' COMMENTS:

Reviewer #2 (Remarks to the Author):

The authors addressed all my questions and comments in the revised manuscript. Therefore I recommend the revised manuscript for publication in Nature Communications.

Reviewer #3 (Remarks to the Author):

In this revised version of the manuscript, the authors have satisfactorily addressed my comments in my previous Referee report.

They have added new materials and sentences/paragraphs to clarify some aspects of their device and to better understand its achievements, and they have corrected typos and language errors throughout the manuscript.

I think this is a beautiful piece of work that deserves publication in Nature Communications. Therefore, I recommend it for publication without any reservation.